# Morphology of travel routes and the organization of cities

Minjin Lee[1], Hugo Barbosa[2], Hyejin Youn [3,4,5,8], Petter Holme [6] & Gourab Ghoshal[2,7]

The city is a complex system that evolves through its inherent social and economic interactions. Mediating the movements of people and resources, urban street networks offer a spatial footprint of these activities. Of particular interest is the interplay between street structure and its functional usage. Here, we study the shape of 472,040 spatiotemporally optimized travel routes in the 92 most populated cities in the world, finding that their collective morphology exhibits a directional bias influenced by the attractive (or repulsive) forces resulting from congestion, accessibility, and travel demand. To capture this, we develop a simple geometric measure, inness, that maps this force field. In particular, cities with common inness patterns cluster together in groups that are correlated with their putative stage of urban development as measured by a series of socio-economic and infrastructural indicators, suggesting a strong connection between urban development, increasing physical connectivity, and diversity of road hierarchies.

[1] Department of Energy Science, Sungkyunkwan University, Suwon 16419, Korea. [2] Department of Physics and Astronomy, University of Rochester, Rochester, NY 14627, USA. [3] Kellogg School of Management at Northwestern University, Evanston, IL 60208, USA. [4] Santa Fe Institute, 1399 Hyde Park Road, Santa Fe, NM 87501, USA. [5] London Mathematical Lab, London WC2N 6DF, UK. [6] Institute of Innovative Research, Tokyo Institute of Technology, Nagatsuta-cho 4259, Midori-ku, Yokohama, Kanagawa 226-8503, Japan. [7] Goergen Institute for Data Science, University of Rochester, Rochester, NY 14627, USA. [8] Northwestern Institute on Complex Systems, Northwestern University, Evanston, IL 60208, USA. Minjin Lee and Hugo Barbosa contributed equally to this work. Correspondence and requests for materials should be addressed to G.G. (email: gghoshal@pas.rochester.edu)

The city is an archetype of a complex system, existing and evolving due to the myriad socio-economic activities of its inhabitants[1–3]. These activities are mediated by the accessibility of urban spaces depending on the city topography and its infrastructural networks[4], a key component of which are the roads. Indeed, different street structures result in varying levels of efficiency, accessibility, and usage of the transportation infrastructure[5–11]. Structural characteristics, therefore, have been of great interest in the literature[12–15] and many variants of structural quantities have been proposed and measured in urban contexts, including the degrees of street junctions[16], lengths of road segments[15], cell areas or shapes delineated by streets[13], anisotropies[14], and network centrality[17,18]. Collectively, these structural properties have uncovered unique characteristics of individual cities as well as demonstrated surprising statistical commonalities manifested as scale invariant patterns across different urban contexts[19–21].

While these studies have shed light on the statistical structure of street networks, there is limited understanding of the interplay between the road structure and its influence on the movement of people and the corresponding flow of socio-economic activity; that is, the connection between urban dynamics and its associated infrastructure[22]. One way to tease out this connection is to examine the sampling of routes, that is an examination of how inhabitants of a city potentially utilize the street infrastructure. While a number of studies have been conducted on the empirical factors behind the choice of routes[23–27], much remains to be done, in particular, understanding the morphological properties of route choices.

Indeed, the morphology of a route is shaped by the embedded spatial pattern of a city (land use and street topology) in association with dynamical factors such as congestion, accessibility, and travel demand, which relate to various attendant socio-economic factors. Analyzing the morphology of routes, therefore, allows us to potentially uncover the complex interactions that are hidden within the coarse-grained spatial pattern of a city. Furthermore, the morphology also encodes the collective property of routes, including their long-range functional effects. For example, a single street, depending on its connectivity and location, can have influence that spans the dynamics across the whole city (Broadway in New York City for instance)[5].

In particular, traffic patterns and the shape of routes have been shown to be determined, among other factors, by two competing forces[24]. On the one hand, one finds an increased tendency of agglomeration of businesses, entertainments, and residential concerns near the urban center, correspondingly leading to a higher density of streets[28,29] and thus attracting traffic and flows toward the interior of the city (positive urban externality). Conversely, this increasing density leads to congestion and increase in travel times (negative urban externality) thus necessitating the need for arterial roads or bypasses along the urban periphery to disperse the congestion at the core. This has the effect of acting as an opposing force, diverting the flow of traffic away from the interior of the city.

Here, we investigate these competing effects through a detailed empirical study of the shape of 472,040 travel routes between origin–destination points in the 92 most populated cities in the globe, representing all six inhabited continents. Each route consists of a series of connected roads; accompanying information on their geographical location, length, and speed limit retrieved from the OpenStreetMap database[30]. We split our analysis between the shortest routes (necessarily constrained by design limitations and city topography) and the fastest routes (representing the effects of traffic and dynamic route sampling), with the former representing aspects of the city morphology, while the latter in some sense representing the dynamics mediated by the morphology (see Methods for details). Specifically, the shortest routes are a function of the bare road geographic structure, while the fastest routes represent the effective geographic structure—a function of the heterogeneous distribution of traffic velocity resulting from varying transportation efficiency and congestion patterns[31–34]. To uncover the functional morphology of these two categories of routes, we define a geometric metric, which we term inness—a function of both the direction and spatial length of routes—that captures the tendency of travel routes to gravitate toward or away from the city center. This metric serves as a proxy for the geographical distribution of attractive forces that may be implicit in the sampling of streets (as reflected in directional bias) and that otherwise cannot be captured by existing measures. Our analysis represents a step toward the very important challenge of determining the spatial distribution of urban land-use and street topology to balance the inherent negative and positive urban externalities that result from rapid urbanization[24].

## Results

**Definition of inness I**. Figure 1 illustrates the forces related to a city's morphological patterns shaped by infrastructural and socio-economic layouts. For the case of a square grid, as shown in Fig. 1a, the shortest routes between any two points at a distance $r$ either correspond trivially to the line connecting them directly, or are degenerate paths that traverse the grid in either direction. Taking the average of the multiple paths cancels any directional bias relative to the center of the grid. Yet, a small perturbation of

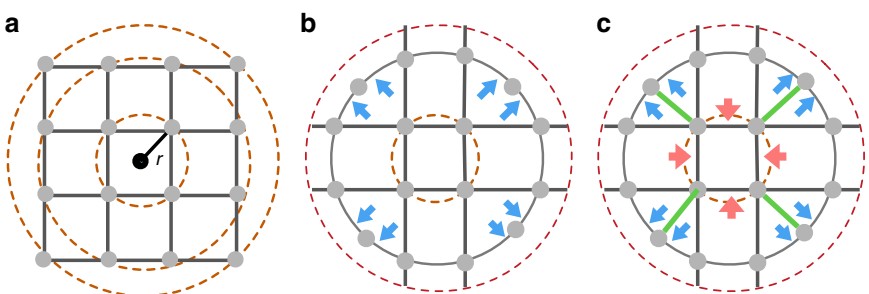

**Fig. 1** Biasing forces found in urban morphology. Three schematic urban street arrangements share similar topological structure, but different geometric layouts resulting in varying dynamics. **a** A grid structure where the shortest paths between points at the same radius show no directional bias. **b** Repulsive forces relative to the origin (marked in blue) emerge as we break the grid symmetry by relocating the four outer points on the inner equidistant ring line. Paths lying on this ring now have the shortest paths that traverse the periphery and avoid the center. **c** Further perturbing the topology by increasing connectivity to the center (marked as four green lines) now leads to shortest paths that go through the center as if an attractive force is present (marked in red)

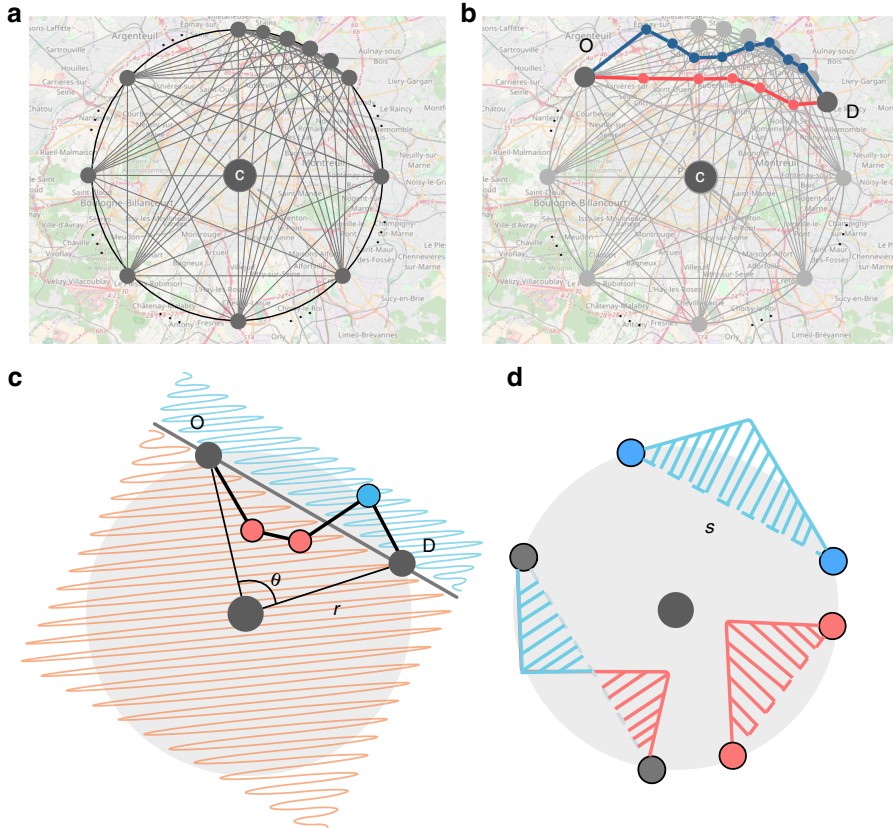

**Fig. 2** Data sampling and definition of inness *I*. **a** Thirty-six origin–destination (OD) pairs (spaced out at intervals of 10°) are assigned along the circumference of circles at a distance of 2, 5, 10, 15, 20, and 30 km from the city center C. **b** For each OD pair, we query the Open Source Map API and collect the shortest routes (red) and the fastest routes (blue) (shown here for a representative OD pair in Paris). **c** A typical OD pair with the straight line connecting them representing the geodesic distance *s*; *r* is the radial distance from the center and *θ* is the angular separation relative to the center. We define the inness (*I*) to be the difference between the inner travel area (polygon delineated by red inner point and straight line) and the outer travel area (polygon delineated by blue outer point and straight line). **d** Three possible route configurations between multiple OD pairs. One with an exclusively outer travel area (blue), one with an exclusively inner travel area (red), and one where there is some combination of both

this regularity can change this neutral feature dramatically, as shown in Fig. 1b, where we shift the four outermost points inward as to place them on the second ring from the center. Points lying on this ring have shortest routes that lie along the periphery, thus introducing a dispersive force away from the center (marked as blue arrows). In Fig. 1c, we further perturb the topology by adding four lines from the outer ring to the inner ring (marked in green) thus increasing connectivity toward the center. Shortest paths between pairs on the outer ring traverse through the inner ring and are curved toward the city center, resulting in an attractive force (represented as red arrows). Beyond this simple example, which is primarily a function of topology and is applicable to shortest paths, other factors are in play such as travel time and route velocity as considered in ref. [24], that will necessarily affect the patterns seen in fastest paths. Furthermore, the illustration assumes a single center of gravity, as it were, whereas such an effect may manifest itself at multiple scales, resulting in cancellation of any measurable force toward a putative city center.

To capture whether such an effect manifests itself at the scale of the city, or is indeed neutral due to the "detuning" at smaller scales, we define a metric called the inness *I*. Figure 2c illustrates how a typical route between any origin–destination (OD) pair can be divided into segments that are directionally biased toward or away from the city center as measured relative to the geodesic distance *s* between the pair. We label points lying closer to the

center than the geodesic inner points while those lying further away are outer points. For example, in the schematic shown in Fig. 2c, points located in the pink shaded area are inner points, and those on the opposite side (shaded blue) are outer points. We define an inner travel area delineated by the polygon of inner points and the geodesic line, to which we assign a positive sign. Conversely, an outer travel area is defined by the geodesic line and the collection of outer points, whose sign is negative. Having adopted this convention, *I* is the difference between the inner area and outer travel areas:

$$I = A_{\text{in}} - A_{\text{out}}, \qquad (1)$$

which can be calculated using the shoelace formula for polygons (see Methods). In Fig. 2d, we show three possible idealizations of a route; one with only outer travel area (blue), one with only inner travel area (red), and one with a mixture of both outer and inner travel areas (combination of blue and red).

The inness of a node in the road network is a result of aggregating characteristics of all possible routes that pass through that point. Indeed, it reflects the structure of the network since it is a metric influenced by topology and network connectivity. However, in addition to this, it captures the geometric aspects of the network since it is a measure based on the curvature of the roads along a route and encodes structural information at both the global and local scales. In that sense, one can consider it to

contain elements of various standard structural network metrics (see Supplementary Note 2 for details and comparison with a series of network metrics). We note that such metrics—particularly those that are global measures such as the betweenness centrality—have been previously used to classify cities. Recent results, however, cast doubt on the efficacy of centrality measures to distinguish between cities[35]. On the other hand, as we will demonstrate, the inness, in addition to being a relatively simple measure, encodes the geometric, infrastructural, geographical, and socioeconomic aspects of urban systems.

**Average inness for shortest and fastest routes**. We begin our analysis by an examination of the qualitative trends of $I$. In Fig. 3a–f, we plot the average inness $\langle I \rangle$ (averaged over the 92 cities) for both the shortest (green curve) and the fastest routes (purple curve) as a function of the angular separation $\theta$, for multiple radii $r$. In the vicinity of the city center around 2–5 km (Fig. 3a, b), we see a neutral trend for the shortest routes ($\langle I \rangle \approx 0$) although fluctuations increase with angular separation ($80° \leq \theta \leq 160°$). The fluctuations anticipate a clear inward bias that emerges at a distance of $r \geq 10$ km (Fig. 3c–f), visible as a pronounced positive peak in $\langle I \rangle$ that grows progressively sharp with increasing $r$. The qualitative trend of $\langle I \rangle$ is indicative of the presence of a core–periphery street network structure present in varying degrees across all cities[36] and suggests that the attractive forces introduced in Fig. 1 tend to manifest themselves at the scale of the entire city, pointing to the existence of an effective center (on an average). Indeed, for fixed $r$, the geodesic distance $s$ between any OD pair is a monotonically increasing function in $\theta$. The longer the distance, the more likely it is for the route to drift toward the center—due to greater connectivity in the center compared to the periphery. This is a possible explanation for the observed inward bias and is indicative of a progressively lower density of streets in the periphery.

The roads in our dataset are organized in a hierarchical fashion consisting of motorways, primary and trunk roads at the top of the hierarchy and residential and service roads being at the lower levels (Supplementary Fig. 2). The shortest paths considered so far are primarily composed of secondary and residential roads (Supplementary Fig. 3); conversely the fastest routes tend to use a smaller subset of the overall network, primarily motorways that are in general major highways with physical divisions separating flows in opposite directions such as freeways and Autobahns (Supplementary Fig. 4). This heterogeneity in the road capacity is bound to introduce differences in the inness profiles of the shortest and fastest routes. Reflecting this, one sees an inward bias for the fast routes emerging around 10 km, but markedly less pronounced than seen for the shortest routes, although the qualitative trend of increasing inward bias with $r$ is maintained. The lower inward bias of faster routes can be explained by the fact that the motorways are typically located in the periphery of cities. Additionally, the angular range of the observed inward bias is lower than that seen for the shortest paths ($45° \leq \theta \leq 120°$) and the fluctuations are significantly larger. This is indicative of the heterogenous spatial distribution of velocity profiles in the primary roads across cities (due to varying levels of infrastructure), coupled with the fact that they are in general longer than secondary roads.

The collective angular and radial dependences of $\langle I \rangle$ are shown as density plots for the shortest and fastest routes in Fig. 3g, h. The monotonic increase of $\langle I \rangle$ with $r$ is apparent in both cases particularly at $r \sim 15$ km (also shown explicitly in Supplementary Fig. 5a). The differences in angular dependence can be clearly seen with the fastest routes having a sharp $\langle I \rangle$ at a lower angular range than the shortest paths. Notable is the absence of any outward bias (negative values of $\langle I \rangle$) at any radial or angular range.

**Dimensionless inness $\hat{I}$**. The observed trends of the inness do not take into account the effects of varying travel areas at different distances from the center, or the variation in urban size across the studied cities. To account for these effects, we note that the travel area (averaged across cities) increases roughly quadratically with the geodesic distance $s$ (Supplementary Fig. 5k), a trend also observed in ref. [23] where the characteristic shapes of city routes had travel areas of $O(s^2)$. Therefore, to account for any bias from variations in travel area within and across cities, we define a rescaled inness

$$\hat{I} = \frac{I}{s^2}. \tag{2}$$

In Fig. 3i, j, we plot $\langle \hat{I} \rangle$ for the shortest and fastest routes finding that the inness effect is robust to potential biases due to length or area of trips. While the qualitative behavior is similar to that seen for $\langle I \rangle$, the inward tendency of routes is present over a broader $r$ and $\theta$ for both the shortest and fastest routes. For instance, inward biases are apparent at distances of 5 km from the city center, an effect suppressed in $\langle I \rangle$ due to the correspondingly smaller travel area. Furthermore, we now find a relatively more homogeneous distribution with a comparatively weaker dependence on $r$ and $\theta$. The trend for $\langle \hat{I} \rangle$ continues to support an average core–periphery structure in the cities we study (and therefore a city center), in combination with a smoothly decreasing density distribution of streets away from this center. The weaker angular dependence, in particular, hints at an isotropic variation in density of street junctions.

The distribution of $\langle \hat{I} \rangle$ is comparatively less homogeneous for fastest routes, primarily due to the predominance of the high-capacity roads (Supplementary Fig. 2) adding, therefore, more variation to the inness profile. Across a wide range of $r$ and $\theta$, $\langle \hat{I} \rangle$ is generally lower than the shortest routes, while there is sharp increase at $15 \leq r \leq 30$ and $40 \leq \theta \leq 100$. This is likely due to some of the motorways being specialized structures such as ring roads or bypasses that serve as attractors for traffic in the city periphery. This is confirmed by plotting the ratio $\langle \hat{I} \rangle_f / \langle \hat{I} \rangle_s$ in Fig. 3k, where one sees a factor of two or more inward bias in the fastest routes as compared to the shortest routes near the city periphery (~25 km).

**Inness distribution for individual cities**. Having examined the properties of average directional biases across urban areas, we now turn our attention to the patterns in individual cities. Indeed, as the fluctuations in Fig. 3a–f show, there is variability in the inness pattern across cities, reflecting the differences in the level of road hierarchies and organization. The composition of the shortest and fastest routes in terms of different road hierarchies varies significantly from city to city. For example, in some cities (Atlanta, Houston, Madrid) the fastest routes tend to be through motorways, whereas in others (Luanda, Kolkata, and Pune) they tend to be composed of primary roads (Supplementary Fig. 4); these differences are likely to affect their respective inness profiles.

To investigate the effect of these differences, we plot each city as a function of the standard deviation and the average of $\hat{I}$ for the shortest paths (Fig. 4a–c). Most cities are in the range $0.0 \leq \langle \hat{I} \rangle \leq 0.08$ with some outliers at both positive and negative values. Ignoring the outliers for the moment, roughly speaking, we identify three regions: low average and low standard deviation (LL), low average and high standard deviation (LH), and high average and high standard deviation (HH). (See Supplementary Figs. 6 and 7 and Supplementary Note 5 for details on individual and outlier cities.) Each city is also colored according to three metrics reflecting infrastructural and geographical features: in 4a,

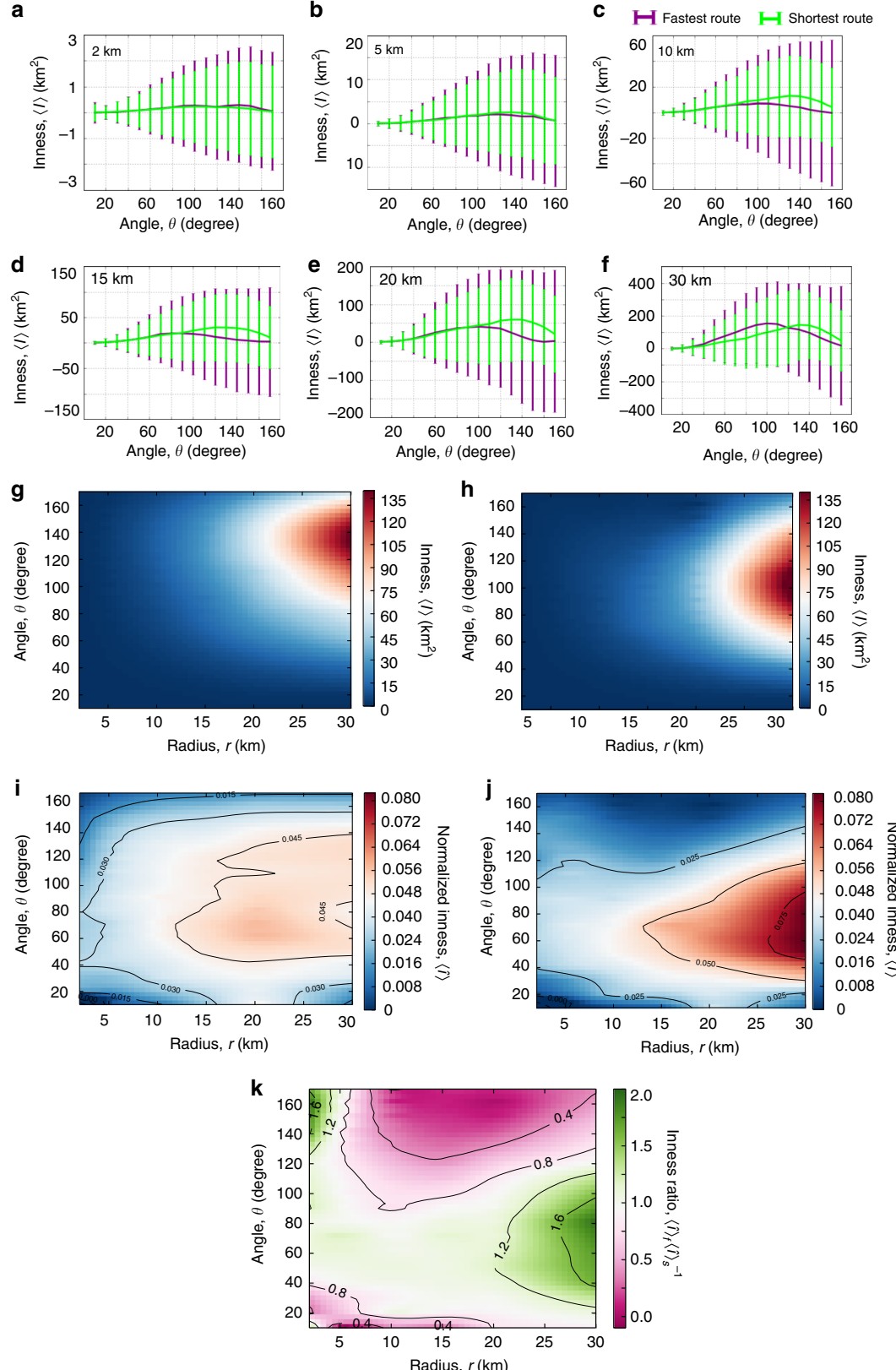

**Fig. 3** The average inness across 92 cities. The average and standard deviation of $l$ as a function of $\theta$ shown for multiple radii $r$ measured from the city center; **a** 2 km, **b** 5 km, **c** 10 km, **d** 15 km, **e** 20 km, and **f** 30 km. The curve for the shortest route is shown in green and the fastest route in purple. The density plot of $\langle l \rangle$ in function of $r$ and $\theta$ for the shortest routes **g** and fastest routes **h**. The normalized or dimensionless inness $\langle \hat{l} \rangle = \langle l/s^2 \rangle$ for the shortest **i** and fastest routes **j**. **k** Ratio of the normalized inness of fastest $\langle \hat{l} \rangle_f$ and shortest routes $\langle \hat{l} \rangle_s$

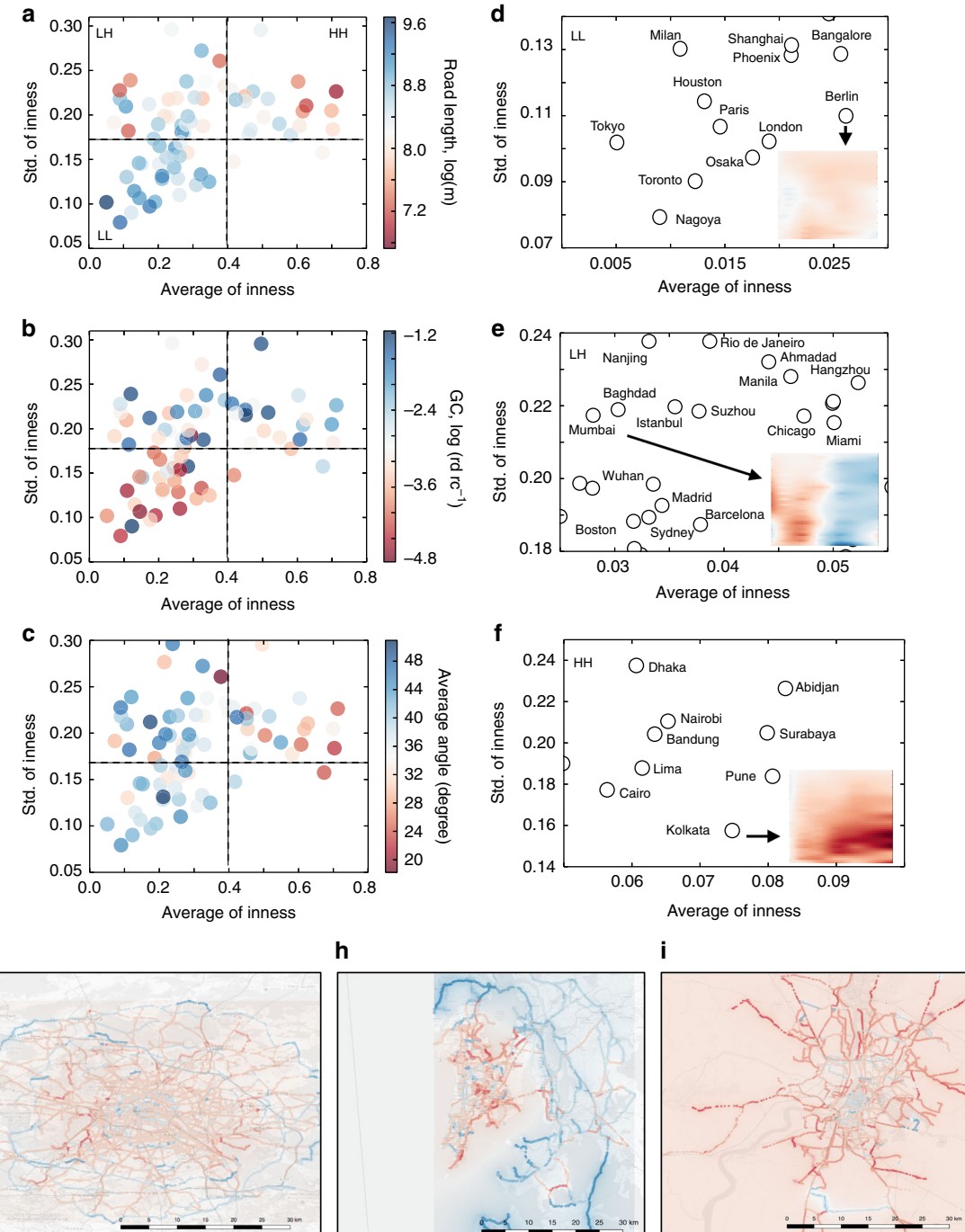

**Fig. 4** The statistics and spatial distribution of the inness for individual cities. The standard deviation plotted in function of average inness for each city. The cities are divided into three groups by their value of average and standard deviation of inness; Low–Low (LL), Low–High (LH), High–High (HH). The color of the points indicates the road length **a**, the level of geographical constraint (GC) **b**, and a measure of peripheral connectivity **c**. We enlarge three zones marked LL, LH, and HH and label the cities explicitly in **d**–**f** as well as display in inset $\hat{I}$, for representative cities in each region (Berlin, Mumbai, Kolkata). In **g**, **h**, and **i** we plot the spatial distribution of $\hat{I}$ projected on to the physical maps for the three representative cities. The color of street intersections corresponds to the average $\hat{I}$ of all routes passing through the intersection, with values in the interval $-0.3 \leq \hat{I} \leq 0.3$ and ranges from blue to red with increasing $\hat{I}$

we show the total length of roads within 30 km from the city center; in Fig. 4b, we show a measure of geographical constraints (GC) that captures the presence and size of barriers such as rivers, coastlines, mountains, or industrial facilities; finally in Fig. 4c, we show a measure of peripheral connectivity acting as a proxy for the presence of ring roads in the city (details for each metric in Supplementary Note 4). In Fig. 4d–f, we show a selection of cities

from each region, along with the density plot of $\hat{I}$ from a representative city shown as inset.

It appears that cities within each region tend to share some common features with respect to these metrics and their inness profiles. Those in the LL group tend to have longer total length of roads, fewer geographic constraints, and strong peripheral connectivity, indicating high levels of infrastructural

development. The inness profile is typically neutral (as can be seen for Berlin), indicating no discernible center of the city to which routes are drawn. On the other hand, cities in the HH group tend to have shorter road lengths, limited connectivity in the periphery (indicating relatively lower levels of infrastructural development), and more geographical constraints than the LL group. These cities also have a markedly positive inness profile (shown for Kolkata) suggesting that the navigability of the city passes through a central core. The LH group appears to show a combination of high and low values in terms of the infrastructural metrics, yet is notable in showing markedly higher geographical constraints than the cities in the other regions. This is reflected in a rather peculiar inness profile which manifests itself positively at short distances, yet is negative at longer ranges.

To investigate these trends further, we plot the spatial distribution of $\hat{I}$ on a geographic map of a representative city from each region. The spatial distribution is generated by considering every intermediate point in a city route, and calculating the average $\hat{I}$ of all routes that pass through this particular location. In Fig. 4g, we show the spatial distribution of $\hat{I}$ for Berlin. Throughout a swathe of Berlin, we find a homogeneous distribution of moderately positive (almost neutral) inness (red), with roads near the city boundary showing a marginally negative inness value (blue). Similar patterns are observed in other large urban agglomerations such as Tokyo and Paris (Supplementary Fig. 10a, c). By and large these cities are large urban areas with advanced infrastructure and strong levels of connectivity.

Next, we focus on the LH group of cities, those with a mix of inward and outward bias in the route patterns. In Fig. 4h, we show the spatial profile for Mumbai which displays two distinct regions with inner and outer bias. The two regions are separated by the Arabian Sea, and connected by (a few) bridges. The left-hand side of the map corresponds to the more densely connected part of Mumbai (its economic center), thus most routes within this region have an inward bias. The appearance of the outward bias in the other region is due to the lack of direct connectivity with the economic center, as the routes have to pass through one of the few bridges that connect the island to the mainland and are thus subject to considerable detour. A similar pattern is seen in other cities in this group, almost all of which have geographic barriers (rivers, seas, hills, mountains) that either spread out across the city, or divide the city into distinct regions. Additionally there are cities in this group with no geographical barriers, but artificial barriers attributing a similar effect on route profiles. A notable example of this is Miami which has a prominent rock-mining industrial site in the Western Miami-Dade County (see Supplementary Fig. 10e, g for this and other examples).

Finally, we examine the spatial distribution of $\hat{I}$ for Kolkata as a city in the HH category, shown in Fig. 4i. It is apparent that the profile is more distinct than that seen for Berlin. There is a hub and spoke-type pattern with the spokes showing high levels of inness, presumably due to its function connecting outer regions to the city center. Indeed, a clear city center is apparent with limited to no connections across the periphery. A similar pattern is seen for other cities in this group (Cairo, Medan, Supplementary Fig. 10i, k). There are at least two possible causes for this: either these cities have a relatively smaller effective urban area (regions of strong connectivity), or they are large urban agglomerations with limited or underdeveloped infrastructure.

**Differences between shortest and fastest routes in cities**. The measured connection of inness profiles with infrastructural indicators suggests that more information can be gleaned by

studying the differences between shortest and fastest routes. Indeed, while the former is more connected to spatial and geographical constraints, the latter shares a more natural connection with developmental indicators. To better quantify this difference, we measure the Pearson correlation coefficient $\rho$ between $\langle\hat{I}\rangle_{\mathrm{f}}$ and $\langle\hat{I}\rangle_{\mathrm{s}}$, for each city, shown in Fig. 5e in increasing order from negative to positive values. Shown as dashed vertical lines are the results of using K-means clustering and Jenks natural breaks optimization to partition the cities into three groups with both methods producing nearly identical divisions. (Alternative clustering approaches revealed similar results, see Supplementary Note 6.)

To investigate this partitioning, we pick a representative city from each end of the spectrum: Berlin with $\rho < 0$ and Mumbai with $\rho \approx 1$. Figure 5a, b show the spatial distribution of $\hat{I}$ in Berlin for the shortest and fastest routes. Unlike the neutral inness seen for shortest routes, fastest routes display a strong outward bias starting from a radial distance of 15 km, which seems to be a consequence of ring-like arterial roads dispersing traffic away from the center. Indeed, this supports our metaphor of competing forces sketched in Fig. 1; the high connectivity of streets in the center of Berlin tends to draw flow toward the city (witnessed by the mildly positive inness profile of the shortest routes) while the faster roads push routes outward. A similar trend is seen for all cities with a negative correlation (Tokyo and Paris shown in Supplementary Fig. 10a–d), with the presence of arterial ring roads (built presumably to alleviate congestion) near the city periphery being the main driver of the differences (in line with higher values of peripheral connectivity measured earlier). A majority of these cities, that are members of partition Type I, correspond to those seen in the LL group in Fig. 4d.

Unlike Berlin, Mumbai exhibits virtually identical profiles of inness between shortest and fastest routes, as seen in Fig. 5c, d, with fewer arterial roads or bypasses that can divert traffic away from the city center. In the case of Mumbai, this is due to salient geographic constraints, but a similar pattern is also seen in other cities like Kolkata, which also lack peripheral roads. Thus, cities that either suffer from some kind of geographic constraint or relatively underdeveloped infrastructure tend to show a higher correlation between the two types of routes. These cities in partition Type III include the majority of cities in the HH group and a few from LH groups in Fig. 4.

Cities with intermediate correlation in partition Type II tend to be those with a profile seen in the LH group in Fig. 4. The behavior seen here seems to be some combination of what drives the trends seen in Type I and Type III cities. As it happens, however, there is the exceptional case of New York. The city shares the same features as Type I cities, i.e., it is a large urban area, with a highly developed infrastructure; yet, there appears to be a strong correlation between the fastest and shortest routes (Supplementary Fig. 14). This is likely due to the unique geography of motorways in the New York metropolitan urban area, which unlike typical Type I cities does not have ring-like motorways in the periphery. Instead, New York consists of a series of radial and grid-like motorways whose overall effect is to cancel any observable directional bias.

Advanced levels of infrastructure are typically reflected in improvements across a variety of socio-economic indices. To examine whether the measured behavior of the inness captures any of this, we consider three socio-economic indicators: the productivity index sourced from the city prosperity index (CPI) created by the UN (http://cpi.unhabitat.org/download-raw-data), the infrastructure development index (also sourced from CPI), and finally the GDP-per-capita sourced from https://www.lloyds.com/cityriskindex/locations. In Fig. 5f–h, we plot these metrics as a function of the correlation coefficient $\rho$ that was used to cluster

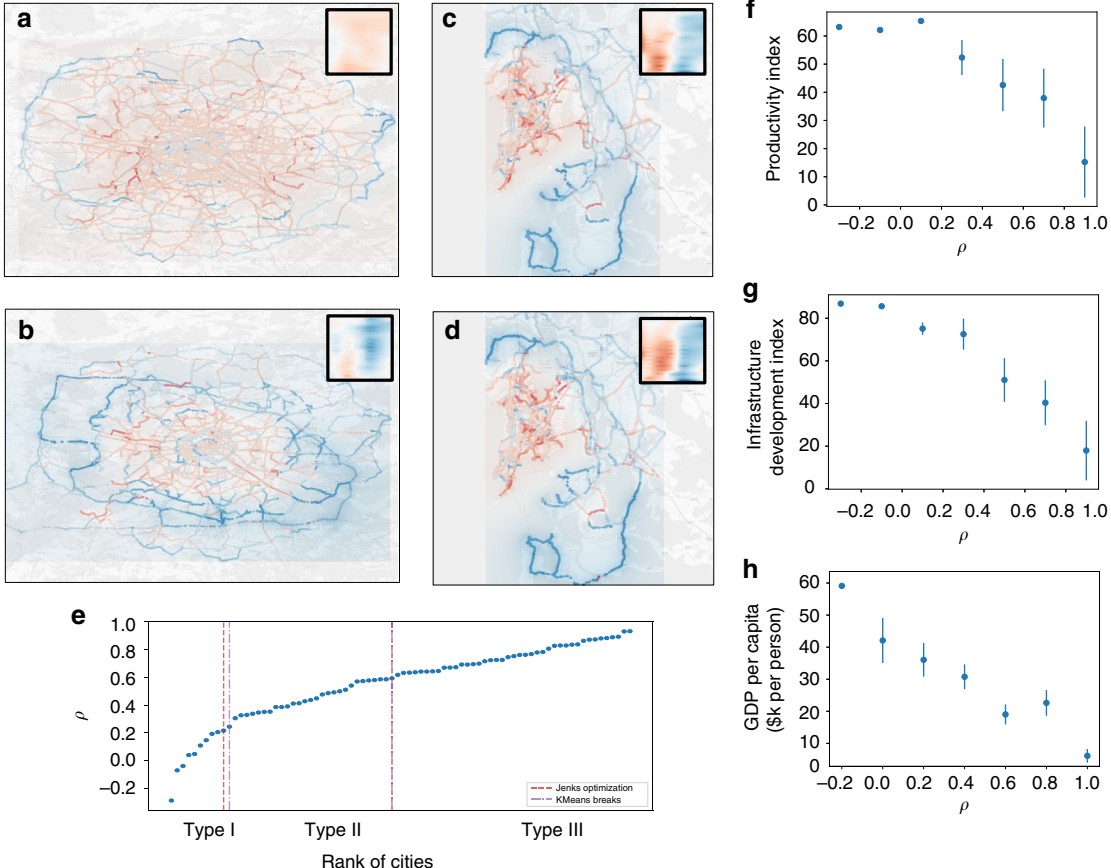

**Fig. 5** Difference between shortest routes and fastest routes. Shortest routes **a** and fastest routes **b** for Berlin. Shortest routes **c** and fastest routes **d** for Mumbai. The insets show the density plots of $\hat{I}$ with the same range as in Fig. 4. **e** Pearson correlation coefficient $\rho$ between the inness patterns of shortest and fastest routes for each city. Cities are categorized into three groups (marked by vertical dashed lines) based on a $K$-means clustering and Jenks natural breaks optimization, conditioned on their level of correlation $\rho$. The names of cities in each type are listed in Supplementary Note 6. Three socio-economic indices, Productivity index **f**, Infrastructure development index **g**, and GDP per capita **h**, are plotted as a function of $\rho$ showing a clear monotonically decreasing trend. Points are averages over cities binned in intervals of 0.2, and bars represent the standard error

the cities. In all three cases, there is a clean monotonic decrease in the indicators with increasing $\rho$, suggesting that inness also encodes information on socio-economic development. A relatively clear pattern emerges whereby the majority of Type I cities are large urban agglomerations with advanced infrastructure and strong socio-economic indicators, Type III cities by and large have comparatively limited infrastructural and socio-economic development, and finally Type II cities share a combination of these features.

## Discussion

The networks of streets and roads are the primary facilitators of movement in urban systems, allowing residents to navigate the different functional components of a city. Since navigability is a key ingredient of socio-economic activity, street networks represent one of the key (if not the most important) infrastructural components. In particular, the utilization of street networks captures the complex interactions between people, and the flow of goods and services in urban systems. However, there is relatively limited understanding of this facet as existing macroscopic or microscopic measures are not able to fully capture its properties and associated effects. Part of the challenge is the limited availability of detailed and high-resolution data of dynamics taking place on such networks, necessitating a choice for investigative studies to be made in terms of granularity or scale. In this manuscript we erred on the side of the latter, and conducted a

systematic mesoscale analysis of street morphology—representing a proxy for the potential dynamics—through the introduction of a metric that we term inness. The inness encapsulates the direction, orientation, and length of routes, thus revealing the morphology of connectivity in street networks, including the implicit infrastructural and socio-economic forces that may inform routing choices.

The average inness pattern points toward the existence of a core–periphery structure across the majority of cities with a high density of streets in the city center with a progressively lower density as one moves toward the periphery. This pattern is particularly seen among Type III cities mostly corresponding to those in Fig. 4f and Supplementary Fig. 10i–l. These happen to be the most numerous in our sample, and therefore dominate the average statistics. Their spatial distribution is characterized by a hub–spoke structure, with Kolkata being an archetype (Fig. 4i), and have a strong correlation between the patterns of fastest and shortest routes. Given that many (but not all) cities in this category are in developing countries (as confirmed by GDP-per-capita and prosperity indices), this feature seems to indicate a relatively underdeveloped infrastructure (confirmed by lower values of infrastructural development index) with the absence of bypasses, highways, or ring-roads that disperses traffic more efficiently.

Interestingly, these qualitative features capture certain elements of the classical hypothesis of central place theory advanced by Christaller[37]. The theory postulates that cities are organized into a

hierarchy of cores that perform specific economic functions depending on their position in the hierarchy[38]. As the core at the highest level of the hierarchy (which is usually located near the city center) has the most diverse and complex economic functions, lower cores that perform less complex functions need to connect to the higher core to meet the demand of some of the activities in the higher core. Furthermore, cores at the lower levels of the hierarchy have minimal interactions between them.

An alternative version advanced by Lösch[39] claims instead that cores at the same level develop specialized and symbiotic functions and thus develop interactions with each other, having the effect of dispersing activity from the center. Indeed, this feature seems to be present in our Type I cities which compose the majority in Fig. 4d and Supplementary Fig. 10a–d. These are predominantly large urban areas with high levels of infrastructural and socio-economic development (Fig. 5f–h). Cities such as Berlin, Paris, and London display a relatively neutral inness trend across locations, pointing toward a uniform density of streets across the city. In particular, a comparison of the spatial distribution of inness for the fastest and shortest routes reveals the presence of ring roads connecting the peripheral areas of the city, and dispersing traffic away from the city center (Fig. 5a, b). Considering one of the purposes of building peripheral roads is facilitating material movement around the city, the observed discrepancy between the shortest and fastest routes might reveal that a developed urban area decouples material (resource) movement from human movement via the introduction of peripheral roads. This is in line with the assertion that cities tend to transform their social and economic functionality into dematerialized operations as they develop[40–42].

A number of cities fall between the spectrum of these two levels of categorization. These are predominantly Type II cities, the majority of which are in the LH group in Fig. 4e and Supplementary Fig. 10e–h. Most (but not all) have geographical or artificial constraints within the city (Mumbai and Rio de Janeiro being notable examples), leading to a mixture of dense and poor connectivity between different locations. Indeed, some have advanced infrastructure (Miami), while others less (Medan) and they also differ in terms of the size of the urban areas. These lie somewhere in between the Christallerian and Löschian classification; although one can take the argument only so far, absent other detailed dynamical information on socio-economic activity and land use pattern, as well as historical data on their evolution.

One must note that inness assumes the existence of a unique city center. Cities that show a strong positive value of inness may thus be considered monocentric in the sense that there is a measurable center through which the majority of routes pass, while those with neutral inness are polycentric, in that there is no such center. Centricity in this case has to be interpreted in the morphological sense reflecting the hierarchical organization and spatial patterns of roads. Cities, in general, however, exist in a continuum between mono and polycentricity, depending on how one defines these terms. Generally, centricity in cities has been measured on the basis of locations of high population density or spatial patterns of land use[43,44], with no dominant quantitative definition for what constitutes the type of centricity[45]. Indeed, the notion of whether cities are monocentric or not depends on the specific feature being investigated. It is notable, however, that some Type I cities considered to be polycentric in terms of our definition are also similarly classified based on completely different metrics such as types of employment and their density patterns[43–45]. Though one must be careful with the analogy, given the limited basis of comparison, inness may be also interpreted as a parsimonious and easily measured metric of centricity in cities, given its simple definition and intuitive interpretation.

In summary, it appears that inness as a simple metric encodes a surprising amount of geometric, infrastructural, and socio-economic information of cities. Indeed, the observed connections between inness, road hierarchies, differences in fast and direct routes, geographical accessibility, centricity in cities, socio-economic indicators, and developmental stage, in combination, provide a rather comprehensive picture of urban organization. The results presented here invite more integrated analysis and interpretation, with respect to existing geographical, historical, morphological, and transport-planning studies for individual cities and their development trajectories.

## Methods

**Sampling routing pairs**. For each of our 92 cities, a city center is defined by referencing the coordinates from latlong.net and the travel routes are sampled according to a choice of OD pairs relative to the center and measured in spherical coordinates (distance from center $r$, and angular separation relative to center $\theta$). To avoid any sample bias, and to systematically investigate the dependence of route morphology on distance from the urban center, we only consider OD pairs at a fixed radius $r$.

Furthermore, at each $r$ we section the circumference of the circle at intervals of 10° for a total of 36 points (with the minimal angular separation chosen to avoid effects of noise). We then vary the radius over the range 2, 5, 10, 20, and finally 30 km (roughly corresponding to a city boundary) and enumerate over all OD pairs by connecting the 36 points at a given radius $r$ for a total of $5 \times \binom{36}{2} = 3150$ total OD pairs.

Finally, we query the OpenStreetMap API for the suggested travel routes connecting each of the pairs. In fact, for a better characterization of the functional features of the systems (e.g., road capacities) and the role of their hierarchical organizations, we obtained two different kinds of routes between all these pairs: the shortest, based on lengths of road segments, and the fastest that accounts for both the length and the travel time based on flow capacity of the roads (i.e., speed limits, number of lanes, etc.). A visual representation of our methodology in segmenting the city is shown in Fig. 2a and typical examples of the shortest and fastest route for a given city is shown in Fig. 2b. (For more details on our data samples, see Supplementary Note 1, Supplementary Tables 1 and 2.)

**Calculation of inness**. The inness $I$ is calculated by summing over the areas of the number of polygons in the route by using the shoelace formula, thus

$$I = \frac{1}{2} \sum_{i=1}^{m} \text{sgn}(i) \left| \sum_{j=1}^{n} \det \begin{pmatrix} x_{i,j} & x_{i,j+1} \\ y_{i,j} & y_{i,j+1} \end{pmatrix} \right|. \quad (3)$$

Here $n$ is the number of vertices of the polygon, $m$ is the total number of polygons in the route, $(x_{i,j}, y_{i,j})$ corresponds to the coordinate of $j$'th vertex of polygon $i$, and $\text{sgn}(i)$ accounts for our adopted convention for inner and outer points.

**Data availability**. All data needed to evaluate the conclusions are present in the paper and/or the Supplementary Materials. Additional data related to this paper may be requested from the authors and are also available at https://github.com/mlee96/inness_research.

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

## Acknowledgements

This work was supported by the US Army Research Office under Agreement Number W911NF-17-1-0127; Basic Science Research Program through the National Research Foundation of Korea funded by the Ministry of Science and ICT with grant No. NRF-2017R1A2B2005957; and the Global Research Network program through the Ministry of Education of the Republic of Korea and the National Research Foundation of Korea with grant No. NRF-2016S1A2A2911945. Map data copyrighted by OpenStreetMap contributors and available from https://www.openstreetmap.org.

## Author contributions

M.L. and P.H. designed the study. M.L. and H.B. implemented the method. M.L., H.B., H.Y., P.H. and G.G. analyzed the results and wrote the manuscript.

## Additional information

**Competing interests:** The authors declare that they have no competing financial interests.

