## [Peer Review File · Nature Communications]

Reviewers' comments:

Reviewer #1 (Remarks to the Author):

In this manuscript, Lee et al. introduce a new metric, the inness, to assess the organization of travels routes respect to the center of the cities. They apply this new metric to a set a cities and find out a classification according to the level of inness of the routes. The manuscript is well written and the results are of general interest since a similar metric can be easily extrapolated to other spatial networks. I recommend thus the publication of the manuscript after the authors have considered the following comments and suggestions.

First concerning the definition of inness, the authors have applied in Equation (3) a normalization factor to take into account the extension of the cities dividing I by s^2 . This allows them to pass from Figures 3B and 3C to 3D and 3E, respectively. The resulting function still shows some dependence on the radius as can be seen in Figure 3D and 3E. However, one could think that the Shortest Path approach should naturally produce the paths with the largest inness since to go from one place to another this protocol will not doubt in crossing the center of the city if it is on the way. An alternative would be to use as baseline the Shortest Path inness and divide the shortest time results by its inness at each angle radius. This could simplify the subsequent discussion of the results.

One of the main results of the manuscript is the classification of cities in three clusters. However, this division as seen in Figure 4A does not seem very solid. It would be necessary to include some quantitative analysis on the clustering division to prevent the impression that this was just a graphic decision.

The cities have a certain history, most naturally start having a clear center and then they develop from there the roads and infrastructures. It is understandable that when more money is available the road network unfolds and renew quicker. Still the main streets normally survive in the expansion plans and it is not necessarily difficult to find in Europe roads such as the via Appia with hundreds or thousands of years of history and still in use. In contrast, in the US the cities are younger and the introduction of the cars substantially modified the urban plans. It would be important to comment about the relation between the history of the cities, the inness and their classification in three types.

The final is established in a sort of confrontation between Christaller's central place theory and Losch's ideas. These theories apply mostly to the use of the city or cities, and how the activity, business, etc, distribute in space. The infrastructure is surely there to satisfy a demand, and it may be related to the activity but it seems hard to believe that the relation is so direct. On one hand, it is

the result of the history of the city and, on the other, in the best case it informs on the mobility needs of the population and indirectly on the potential location of the activity centers. It seems difficult to accept that it says something on the centers hierarchy or on the kind of relation between them.

Reviewer #2 (Remarks to the Author):

0. This paper devises a metric called 'inness' that reflects differences in how travel paths between locations in a city relate to their distance from a nominated central point; and performs analyses relating the shortest paths and fastest paths between sampled points, for a range of cities.

The analyses are used to interpret differences between cities based on their street network structure. The paper generates systematic empirical results for over 400,000 hypothetical travel paths, for the road networks of 92 cities; and a number of interesting analytic maps and charts are produced.

While the paper appears to be robust in its analysis, as far as it goes, and well written and appropriately illustrated (in accompanying the text, as far as it goes), there are three main concerns.

1. First, while the paper has some interesting interpretations, it is not clear to what extent 'inness' is a significant metric or not. While it is claimed to be a 'comprehensive' measure, it's not clear in what way this is any more comprehensive than any of a number of other network metrics.

Indeed, there is no direct comparison with any other network metrics, and so it is not possible to determine if inness is especially meaningful or useful or not. Any number of other network metrics would presumably allow similar kinds of interpretation of a set of cities (dividing them into classes, and allowing commentary on the degree to which they show different kinds of clustering, connectivity, centrality, and so on).

As it stands, the paper admits (p12) that inness does not in fact distinguish between presumed monocentric and polycentric cities; while there is no example given of where inness does distinguish some urban property that existing measures demonstrably fail to do. This of itself does not deny the

potential interest and insight to be gained from the paper, but it makes its claims less than wholly convincing. Ideally the paper would give some sort of comparison with other network metrics, and comment on the relative significance of inness.

2. Second, the paper's assumptions about centrality, monocentricity and polycentricity are not adequately articulated. For a start, the method seems to be reliant on the identification of a single central point, as if implying there are forces of attraction to (or repulsion from) the centre. But surely attraction exists (to a greater or lesser extent) between any and all points in a city, and only in a predominantly monocentric city would 'the' centre be a meaningful reference point for basing the concept of inness? In a polycentric city region, wouldn't the inherent 'monocentricity' of the inness measure mitigate against its meaningfulness? At the very least, some commentary on this potential limitation would be highly desirable.

Secondly, although both monocentricity and polycentricity are alluded to, there is almost no discussion of what is meant by these terms. Depending on how those are defined – and the scale of resolution employed – almost any city could be defined as either monocentric or polycentric. Therefore it would be desirable to have more discussion of, at least, what the paper means by these terms, and ideally more explicit demonstration of the relative monocentricity or polycentricity of the cities discussed.

Finally, the paper refers to forces (i.e. as if centripetal and centrifugal forces act to stimulate movement) but there is inadequate discussion and justification of this idea. There are arguments against interpreting 'opposing forces' at play, since the apparent centrifugality can be interpreted as simply centripetality 'retuned' at a different scale.

3. Third, the paper is to varying extent misleading in terms of what it is about, and overstates its claims as to what it achieves (this is over and above the significance of inness).

For a start, the paper refers in its title to 'urban socioeconomic patterns', but this is misleading since no socioeconomic patterns are demonstrated. The title and paper also refer to 'travel routes', and while this term (sometimes ambiguously used in different contexts) can be reasonably equated with travel paths (i.e. potential paths of movement through the road network), this is potentially misleading since there are no actual travel patterns studied.

In the abstract (and elsewhere) there is reference to 'functional usage (i.e. the movement pattern of people and resources)' but, again, the present paper has no analysis of actual movements of people or resources. Therefore the impression that this paper does something that others fail to do is

misleading, especially as there are myriad studies of road networks that do address 'functional usage' (in particular, the whole sub-field of space syntax has decades of work relating street network structure to pedestrian and vehicular movement, which is not referenced here).

The abstract also refers to 'diversity of road hierarchies' but the main body of the paper does not directly refer to road hierarchy (another sub-field whose literature is by-passed), other than the acknowledgement that some routes will enable faster speeds than others. There is more direct treatment in the Supplementary Material, but this is not fully explained (whether in terms of speed, capacity, transport mode, or 'diversity of road hierarchies') and this would seem to deserve more discussion, to allow more confidence in the meaning and significance of the results.

Finally, in the discussion it is claimed that 'More than the physical layout, it is the sampling of street networks that serves as a true fingerprint...' – however it is not clear what 'true fingerprint' means, in an arena when surely different metrics tell us different things – all 'true' but with different meaning or significance – about a complex system. And while the paper claims to be 'revealing the morphology of connectivity' it would seem more appropriate to say it is revealing some particular aspect(s) of the morphology of connectivity; while the 'distribution of implicit socioeconomic forces' cannot be comfortably accepted, since no socioeconomic data is considered in the study.

The claim that the study provides 'evidence of this evolution' (of spatial and functional expansion) seems overstated since the study does not track such evolution, and the claim seems to be no more than one can read into the map of any city some aspects of its historical development. All of this is not to deny the value of a detailed and impressively assembled geographical study of network routes/paths, but the overstatement of the socioeconomic and usage angles (from the title onwards) is unhelpful and ultimately counter-productive as it if anything casts doubt on the rest of the paper.

4. In conclusion: the reviewer guidelines for reviewers state:

“Setting out the arguments for and against publication is often more helpful to the editors than a direct recommendation one way or the other.”

In this case the grounds for publication are:

- Demonstration of a novel measure that sheds new light on an aspect of cities' road networks
- Robust empirical analysis based on a large number of cities across the world

The grounds against publication are:

- The significance of this single measure does not yet seem clear – it could be arguably a rather specialist ‘geographical’ measure of interest primarily to other street network structure analysts. The paper could be strengthened by more engagement with previous literature, both on alternative network metrics, and on interpretations of cities’ road networks more generally.
- There are some doubts about the adequacy of the treatment of monocentricity and polycentricity, and centrifugal and centripetal mechanisms – this could be rectified by additional discussion, with reference to existing literature.
- There are reservations about the claims being made that seem overstated – this could be rectified by more accurate, modest framing; but this could have knock-on effects for the claimed significance of the paper, and its suitability for the present journal.

Reviewer #1 (Remarks to the Author):

In this manuscript, Lee et al. introduce a new metric, the inness, to assess the organization of travels routes respect to the center of the cities. They apply this new metric to a set a cities and find out a classification according to the level of inness of the routes. The manuscript is well written and the results are of general interest since a similar metric can be easily extrapolated to other spatial networks. I recommend thus the publication of the manuscript after the authors have considered the following comments and suggestions.

We thank the referee for the kind words and are delighted that they support the publication of the manuscript as well as making the observation that the metric is portable to other spatial networks.

First concerning the definition of inness, the authors have applied in Equation (3) a normalization factor to take into account the extension of the cities dividing I by s^2 . This allows them to pass from Figures 3B and 3C to 3D and 3E, respectively. The resulting function still shows some dependence on the radius as can be seen in Figure 3D and 3E. However, one could think that the Shortest Path approach should naturally produce the paths with the largest inness since to go from one place to another this protocol will not doubt in crossing the center of the city if it is on the way. An alternative would be to use as baseline the Shortest Path inness and divide the shortest time results by its inness at each angle radius. This could simplify the subsequent discussion of the results.

This is an excellent observation and we were remiss in not including this and discussing the point in a clearer fashion. While part of the radial dependence is suppressed by dividing out by the area of the travel paths, artifacts remain because the relation is quadratic only to first order. The situation is more complicated when considering the fastest paths, since the radial dependence only manifests itself near the periphery and is suppressed, or near-absent, towards the city center, hence the difference in profiles between the two. Dividing the fastest path inness by the shortest path inness indeed reveals the regions in which one or the other is more dominant. We have included this new figure as a part of Figure 3 (3F) which now clearly shows that the fastest paths are inward only near the periphery (by a factor of 2 or more than the shortest paths) and the shortest paths are more inwards towards the center. This makes the distinction between the profiles much clearer and we thank the referee for the great suggestion.

One of the main results of the manuscript is the classification of cities in three clusters. However, this division as seen in Figure 4A does not seem very solid. It would be necessary to include some quantitative analysis on the clustering division to prevent the impression that this was just a graphic decision.

Indeed, and we were lacking in both the presentation of Figure 4 and confusing the issue by giving the impression that we were clustering the cities based on that figure. Actually, the city clustering is done only in Figure 5 based on the correlation between the shortest and fastest paths as the determining parameter. Our presentation on this was unnecessarily clumsy. To remedy this, we did the following things:

- We have now substantially revised and relabeled Figure 4 and identified regions in the figure for what they are and nothing more. Cities with low average inness and standard deviation (LL), those with high average inness and standard deviation (HH) and finally those with low average inness and high standard deviation (LH). At this level, there is no clustering, or claim of any division.
- Next, we introduce three separate quantitative metrics (road length, level of geographic constraints, and levels of peripheral connectivity, i.e presence of ring-roads) described in detail in Section S5.1. In three separate figures of the standard deviation vs average inness, we color the cities according to the level of each of these measures. In conjunction with the earlier observation that cities within each region shared a distinct inness profile, we also show that they share common features based on these three metrics. In particular LL cities have longer roads, greater peripheral connectivity and vanishing geographic constraints. Additionally, they have a neutral inness profile. Cities in the HH region have shorter road lengths, stronger geographic constraints, limited-to-no peripheral connectivity and a markedly strong inness profile. Finally, cities in the LH region share a mixture of features, but are marked in particular by strong geographic constraints and a peculiar inness profile with inward tendencies at short ranges and outer detour at long range. At this stage, there is still no mention of clustering, but indications that cities within each region share strong common features across a variety of metrics. This is described in detail in pages 9-11.
- In Figure 5, we plot the Pearson correlation between the inness of the shortest and fastest routes finding a monotonic increase, with LL cities having a negative correlation, HH cities having a strong positive correlation, and LH cities lying in an intermediate range. At this stage, we apply a K-means clustering conditioned on the Pearson correlation coefficient, finding that the best division is into 3 clusters which we term Type I, II and III. We also test the robustness of this division by using Spearman's rank order correlation and the Kendall rank correlation finding near identical results. Additionally, combining all three of these measures into a hierarchical clustering method revealed that the division into 3 clusters is a reasonable partitioning based on the within-clusters sum of squared deviations (WCSS). The details of this are now shown in Section S6.1 in the SI. Only at this point, do we make the observation that by-and-large LL cities correspond to Type I, HH to Type III and LH to Type II. This is described in detail in pages 11-13.
- Finally, we note that while we referred to socioeconomic factors in various parts of the earlier manuscript, we did not explicitly provide any evidence of their connection with inness. Given the new evidence that inness seems to be an indicator of infrastructural development, and given the connection between improved infrastructure and socioeconomic development, we decided to test this out by plotting in Figure 5F-H three composite socioeconomic indicators (prosperity index, infrastructural development index, and gdp-per-capita, details in Section S6.2) as a function of the Pearson correlation coefficient that we used to cluster the cities. Remarkably in all three cases, we find a clear *significant* monotonic decrease of these indicators in function of the correlation coefficient, indicating that the clusters also contain information on the level of socioeconomic development of the member cities. Indeed, a relatively clear pattern emerges whereby the majority of Type I cities are large urban agglomerations with advanced infrastructure and strong socioeconomic in-

dicators, Type III cities by and large have comparatively limited infrastructural and socioeconomic development, and finally Type II cities share a mixture of these features.

We believe in combination these series of infrastructural, geographic and socioeconomic measures provide exceptionally strong evidence for inness being a novel and intuitive metric that captures the level of organization in cities. This is reflected in our new, and we believe, more accurate title *Morphology of travel routes and the organization of cities*.

The cities have a certain history, most naturally start having a clear center and then they develop from there the roads and infrastructures. It is understandable that when more money is available the road network unfolds and renew quicker. Still the main streets normally survive in the expansion plans and it is not necessarily difficult to find in Europe roads such as the via Appia with hundreds or thousands of years of history and still in use. In contrast, in the US the cities are younger and the introduction of the cars substantially modified the urban plans. It would be important to comment about the relation between the history of the cities, the inness and their classification in three types.

This is an excellent point and something we neglected to discuss in our manuscript. However, it seems that inness is tied to the level of socioeconomic and infrastructural development of cities and says very little about age. Indeed, the correlation with age and how developed cities are is unclear. In addition to the European and American examples, one may consider the city of Varanasi in India, which has been continuously inhabited since 1500 BCE (making it 3500 years old!) yet has dilapidated and broken infrastructure by all measures. Staying in the same country, Chandigarh, which was designed by Le Corbusier in the 1940's has significantly more advanced infrastructure. In any case, to say something substantial, one would need detailed longitudinal data on a variety of cities from across the world, to make a clear connection between age and development. Given that we don't have such data, we feel it best to not comment on this aspect. We believe part of the problem is our liberal (and unclear) use of the word "mature" in the earlier version of the text, giving the impression that we are commenting upon the history and age of the city. We have now excised all such ambiguous words from the manuscript and have made the connection between inness and our measured metrics clear.

The final is established in a sort of confrontation between Christaller's central place theory and Losch's ideas. These theories apply mostly to the use of the city or cities, and how the activity, business, etc, distribute in space. The infrastructure is surely there to satisfy a demand, and it may be related to the activity but it seems hard to believe that the relation is so direct. On one hand, it is the result of the history of the city and, on the other, in the best case it informs on the mobility needs of the population and indirectly on the potential location of the activity centers. It seems difficult to accept that it says something on the centers hierarchy or on the kind of relation between them.

We agree. We took the analogy too far, and absent any information on land use, spatial hierarchy, other modes of transportation and most importantly, longitudinal history of a particular city, the connection is only superficial at best. Nevertheless, our new results suggest that it does capture certain elements of

these classifications. To reflect this, we have significantly toned down the language on the connection between central place theory and our clusters, and have made the necessary disclaimers. The discussion begins on page 14 second to last paragraph with

Interestingly these qualitative features capture *certain elements* of the classical hypothesis of central place theory advanced by Christaller...

and ends on page 15 second paragraph with

These lie somewhere in between the Christallerian and Löschian classification, although one can take the argument only so far, absent other detailed dynamical information on socioeconomic activity as well as historical data on their evolution.

In addition, we have removed any reference to the connection between the evolution of cities and their place in the cluster. Finally, we have rewritten, reorganized and shortened the manuscript to improve its readability and have ensure the discussion accurately reflects the presented results.

In summary, we thank the referee for the constructive criticism and the excellent suggestion, the incorporation of which has greatly improved the manuscript. We hope that it is now suitable for publication.

Reviewer #2 (Remarks to the Author):

In this case the grounds for publication are:

- *Demonstration of a novel measure that sheds new light on an aspect of cities' road networks*
- *Robust empirical analysis based on a large number of cities across the world*

While we appreciate the criticism afforded by the referee, we are encouraged by the positive comments on the manuscript and are grateful for their outlining a path to publication.

First, while the paper has some interesting interpretations, it is not clear to what extent 'inness' is a significant metric or not. While it is claimed to be a 'comprehensive' measure, it's not clear in what way this is any more comprehensive than any of a number of other network metrics. Indeed, there is no direct comparison with any other network metrics, and so it is not possible to determine if inness is especially meaningful or useful or not.

This is a very valid point and we were remiss in not including a comparison between inness and other established network based metrics. Indeed, this is a somewhat fundamental step and we thank the referee for pointing out this crucial oversight. To remedy this, on page 5 last paragraph we add:

The inness of a node in the road network is a result of aggregating characteristics of all possible routes that pass through that point. Indeed, it reflects the structure of the network since it is a metric influenced by topology and network connectivity. However, in addition to this, it captures geometric aspects of the network since it is a measure based on the curvature of the roads along a route and encodes structural information at both the global and local scales. In that sense, one can consider it to contain elements of various standard structural network metrics (see Sec. S2 for details and comparison with a series of network metrics).

One of the primary usage of network metrics in the context of cities has been various flavors of centrality. In Section S2 we provide a comparison of the closeness, eccentricity, degree and betweenness centralities with inness for a variety of cities, demonstrating that the information encoded in each of them are quite different from that seen for inness, providing detailed examples of this in three different major cities. This is not particularly surprising. While the network metrics are primary structural, the inness in addition to having structural dependencies, contains information of the geometry of routes, and in the case of fastest routes “quenched” information on the dynamics occurring in the streets, none of which are captured by any of the centrality metrics. Given that street networks are planar, other typical network measures such as the degree distribution, clustering coefficient and other n-point correlation functions are trivially useless, given the strong constraint of planarity making the degree distribution a tightly peaked function.

Any number of other network metrics would presumably allow similar kinds of interpretation of a set of cities (dividing them into classes, and allowing commentary on the degree to which they show different kinds of clustering, connectivity, centrality, and so on).

This is somewhat speculative. Indeed, network metrics can be used to cluster cities and show differences or similarities, but once again the truly meaningful (structural) ones are those that capture global information such as centralities. Indeed, one of the oft used ones is the betweenness centrality (BC). It may interest the referee to know, that recent results have cast doubt on the use of the betweenness centrality to classify cities in any sense. In particular a paper written by two of the co-authors and Marc Barthélemy, conclusively demonstrate (empirically, analytically and by simulations of various random graph models) that the BC distribution for planar graphs (which street networks are) is an invariant quantity (after a proper rescaling). That is to say the distribution is the same for all cities irrespective of geography, location, age etc. In fact, an analysis of 200 years of historical data of Paris, shows that nothing changes in terms of the BC distribution. Details here: <https://arxiv.org/abs/1709.05718>

By contrast, inness can clearly be used to make statements about similarities and differences between cities, and of course a number of other (not-necessarily network) metrics can as well, however, we will make the case below, that it contains a surprising amount of geometric, infrastructural and signatures of socioeconomic information encoded in such a simple measure.

The paper refers to forces (i.e. as if centripetal and centrifugal forces act to stimulate movement) but there is inadequate discussion and justification of this idea. There are arguments against interpreting 'opposing forces' at play, since the apparent centrifugality can be interpreted as simply centripetality 'retuned' at a different scale.

We agree that our discussion on forces was rather too simplistic. In particular we were inadequate in explaining cases where such an effect is likely to be suppressed or detuned as it were. Furthermore, we did not mention here anything about effects apart from geometry. We thank the referee for pointing out this limitation and have made the following refinements. On page 3, last paragraph, we add:

Fig. 2 illustrates schematics of the forces hidden in the city's morphological patterns, shaping, and shaped by infrastructure and socioeconomic layouts. For the case of a square grid, as shown in Fig. 2A, shortest routes between any two points at a distance r either correspond trivially to the line connecting them directly, or are degenerate paths that traverse the grid in either direction. Taking the average of the multiple paths cancels any directional bias relative to the center of the grid. Yet, a small perturbation of this regularity can change this neutral feature dramatically, as shown in Fig. 2B, where we shift the four outermost points inwards as to place them on the second ring from the center. Points lying on this ring have shortest routes that lie along the periphery, thus introducing a dispersive force away from the center (marked as blue arrows). In Fig. 2C, we further perturb the topology by adding four lines from the outer ring to the inner ring (marked in green) thus increasing connectivity towards the center. Shortest paths between pairs on the outer ring traverse through the inner ring and are curved towards the city center, resulting in an attractive force (represented as red arrows). Beyond this simple example, which is primarily a function of topology and is applicable to shortest paths, other factors are in play such as travel time and route velocity as considered in [24] that will necessarily affect the patterns seen in fastest paths. Furthermore, the illustration assumes a single center of gravity, as it were, whereas such effects may manifest itself at multiple scales, resulting in cancellation of any measurable force towards a putative city center.

To capture whether such an effect manifests itself at the scale of the city, or is indeed neutral due to the "detuning" at smaller scales, we define a metric called the Inness I.

We are hopeful that this makes the point clearer. In particular, it makes the distinction between those cities where such effects are not detuned (i.e. the majority of cities that show strong positive inness and therefore the existence of a center or core in terms of navigability) and those where we do not see such an effect, implying that no such center exists. We also make clear what we mean by a center, see below.

As it stands, the paper admits (p12) that inness does not in fact distinguish between presumed monocentric and polycentric cities...Second, the paper's assumptions about centrality, monocentricity and polycentricity are not adequately articulated. For a start, the method seems to be reliant on the identification of a single central point, as if implying there are forces of attraction to (or repulsion from) the centre. But surely attraction exists (to a greater or lesser extent) between any and all points in a city, and only

in a predominantly monocentric city would 'the' centre be a meaningful reference point for basing the concept of inness? In a polycentric city region, wouldn't the inherent 'monocentricity' of the inness measure mitigate against its meaningfulness? At the very least, some commentary on this potential limitation would be highly desirable. Secondly, although both monocentricity and polycentricity are alluded to, there is almost no discussion of what is meant by these terms. Depending on how those are defined – and the scale of resolution employed – almost any city could be defined as either monocentric or polycentric. Therefore it would be desirable to have more discussion of, at least, what the paper means by these terms, and ideally more explicit demonstration of the relative monocentricity or polycentricity of the cities discussed.

We agree that we were not very clear on this point and did not make reference to existing literature on centrality. Given that monocentric and polycentric organization (at least in the established sense) is not the main point of this paper, we removed all references to the same in the main body of the manuscript. Instead we added a discussion on this in the last section to put our findings in context. On Page 15, third paragraph:

One must note that inness assumes the existence of a unique city center. Cities that show a strong positive value of inness, may thus be considered monocentric in the sense that there is a measurable center through which the majority of routes pass; while those with neutral inness are polycentric, in that there is no such center. Centricity in this case, has to be interpreted in the morphological sense reflecting the hierarchical organization and spatial patterns of roads. Cities, in general, however, exist in a continuum between mono and polycentricity, depending on how one defines these terms. Generally, centricity in cities has been measured on the basis of locations of high population density or spatial patterns of land use [45, 46] with no dominant quantitative definition for what constitutes the type of centricity [47]. Indeed the notion of whether cities are monocentric or not, depends on the specific feature being investigated. It is notable, however, that some Type I cities considered to be polycentric in terms of our definition, are also similarly classified based on completely different metrics such as types of employment and their density patterns [45-47]. Though one must be careful with the analogy, given the limited basis of comparison, inness may be also interpreted as a parsimonious, and easily measured metric of centricity in cities, given its simple definition and intuitive interpretation.

The paper is to varying extent misleading in terms of what it is about, and overstates its claims as to what it achieves (this is over and above the significance of inness). For a start, the paper refers in its title to 'urban socioeconomic patterns', but this is misleading since no socioeconomic patterns are demonstrated.

We acknowledge that the title of the paper was a bit misleading, since the referee correctly states that no connection between socioeconomic patterns and inness is developed. The first is an easy fix, we changed the title to more accurately reflect the results: *Morphology of travel routes and the organization of cities*. Furthermore, we now include some rather compelling evidence of the connection between inness and various indicators of socioeconomic development. However rather than just stating that result

in isolation, we think it's important to provide the sequence of reasoning that led us to our conclusion. So to provide context and indeed to *clarify* what this paper is about here are the relevant steps

- We have substantially revised and relabeled Figure 4 and identified regions in the figure for what they are and nothing more. Cities with low average inness and standard deviation (LL), those with high average inness and standard deviation (HH) and finally those with low average inness and high standard deviation (LH). At this level, there is no clustering, or claim of any division.
- Next we introduce three separate quantitative metrics (road length, level of geographic constraints, and levels of peripheral connectivity, i.e. presence of ring-roads) described in detail in Section S5.1. In three separate figures of the standard deviation vs average inness, we color the cities according to the level of each of these measures. In conjunction with the earlier observation that cities within each region shared a distinct inness profile, we also show that they share common features based on these three metrics. In particular LL cities have longer roads, greater peripheral connectivity and vanishing geographic constraints. Additionally, they have a neutral inness profile. Cities in the HH region have shorter road lengths, stronger geographic constraints, limited-to-no peripheral connectivity and a markedly strong inness profile. Finally, cities in the LH region share a mixture of features, but are marked in particular by strong geographic constraints and a peculiar inness profile with inward tendencies at short ranges and outer detour at long range. At this stage, there is still no mention of clustering, but indications that cities within each region share strong common features across a variety of metrics. This is described in detail in pages 9-11.
- In Figure 5 we plot the Pearson correlation between the inness of the shortest and fastest routes finding a monotonic increase, with LL cities having a negative correlation, HH cities having a strong positive correlation, and LH cities lying in an intermediate range. At this stage, we apply a K-means clustering conditioned on the Pearson correlation coefficient, finding that the best division is into 3 clusters which we term Type I, II and III. We also test the robustness of this division by using Spearman's rank order correlation and the Kendall rank correlation finding near identical results. Additionally, combining all three of these measures into a hierarchical clustering method revealed that the division into 3 clusters is a reasonable partitioning based on the within-clusters sum of squared deviations (WCSS). The details of this are now shown in Section S6.1 in the SI. Only at this point, do we make the observation that by-and-large LL cities correspond to Type I, HH to Type III and LH to Type II. This is described in detail in pages 11-13.
- Finally, we note that while we referred to socioeconomic factors in various parts of the earlier manuscript, we did not explicitly provide any evidence of their connection with inness. Given the new evidence that inness seems to be an indicator of aspects of infrastructure, and given the connection between improved infrastructure and socioeconomic development, we decided to test this out by plotting in Figure 5F-H three composite socioeconomic indicators (prosperity index, infrastructural development index, and gdp-per-capita, details in Section S6.2) as a function of the Pearson correlation coefficient that we used to cluster the cities. Remarkably in all three cases, we find a clear *significant* monotonic decrease of these indicators in function of the correlation coefficient, indicating that the clusters also contain information on the level of socioeconomic develop-

ment of the member cities. Indeed, a relatively clear pattern emerges whereby the majority of Type I cities are large urban agglomerations with advanced infrastructure and strong socioeconomic indicators, Type III cities by and large have comparatively limited infrastructural and socioeconomic development, and finally Type II cities share a mixture of these features.

We believe in combination these series of infrastructural, geographic and socioeconomic measures provide exceptionally strong evidence for inness being a novel and intuitive metric that captures important aspects of the level of organization in cities.

The title and paper also refer to ‘travel routes’, and while this term (sometimes ambiguously used in different contexts) can be reasonably equated with travel paths (i.e. potential paths of movement through the road network), this is potentially misleading since there are no actual travel patterns studied.

We believe that this is perhaps a bit “making too fine a point on the matter” as it were. Anyone who reads our paper will have a reasonable idea what we mean here. We decided to continue to use travel routes, since by definition routes have to be a subset of all possible paths which is of course, as correctly pointed out is what we study. We have modified the language throughout the manuscript to reflect this.

In the abstract (and elsewhere) there is reference to ‘functional usage (i.e. the movement pattern of people and resources)’ but, again, the present paper has no analysis of actual movements of people or resources... While the ‘distribution of implicit socioeconomic forces’ cannot be comfortably accepted, since no socioeconomic data is considered in the study.

Indeed, this is true. We do not study any dynamical aspects in this paper. It is of course reasonable to motivate the importance of street networks in the sense that they do mediate the movement of people and resources. We addressed this (potential ambiguity) the following ways. We changed the language to say that we are studying hypothetical or potential movement of people and resources. In the abstract

In comparison, relatively limited attention has been devoted to the interplay between street structure and its functional usage, i.e., the movement patterns of people and resources. To address this, we study the shape of 472,040 spatiotemporally optimized travel routes in the 92 most populated cities in the world. The routes are sampled in a geographically unbiased way such that their properties can be mapped on to each city, with their summary statistics capturing mesoscale connectivity patterns representing the complete space of possible movement in cities.

And in the discussion section we add on page 13 last paragraph:

Networks of streets and roads are the primary facilitators of movement in urban systems, allowing residents to navigate the different functional components of a city. Since navigability is a key ingredient of socioeconomic activity, street networks represent one of the key (if not the most important) infrastructural components. In particular the utilization of street networks captures the complex interactions be-

tween people, and the flow of goods and services in urban systems. However, there is relatively limited understanding of this facet as existing macroscopic or microscopic measures are not able to fully capture its properties and associated effects. Part of the challenge is the limited availability of detailed and high-resolution data of dynamics taking place on such networks, necessitating a choice for investigative studies to be made in terms of granularity or scale. In this manuscript we erred on the side of the latter, and conducted a systematic mesoscale analysis of street morphology---representing a proxy for the potential dynamics---through the introduction of a novel metric that we term inness. The inness encapsulates the direction, orientation and length of routes, thus revealing the morphology of connectivity in street networks, including the implicit infrastructural and socioeconomic forces that may inform routing choices.

We believe this is an accurate reflection of what we present in this manuscript (see our new results on infrastructural and socioeconomic indicators).

There are myriad studies of road networks that do address ‘functional usage’ (in particular, the whole sub-field of space syntax has decades of work relating street network structure to pedestrian and vehicular movement, which is not referenced here).

We are well aware of the space syntax literature. It is indeed an under-cited field among our colleagues in computational social science and urban data science. We didn’t immediately see the relevance of including it in the initial version of the manuscript, but we have reconsidered and include a representative paper ([27] in the main manuscript and labeled [1] here in the response letter).

We are sure the reviewer agrees that space syntax, as originally conceived by Hillier and Hanson [2], has a rather different scope than this paper. This main line of space syntax theory derives from assumptions about visual information processing of pedestrians. It hinges on the premise that the agents are continuously observing, and being influenced by, their environment. Its ultimate purpose is to create tools for urban planners. In our work, agents are not active in that sense—just car travelers along shortest or fastest routes from source to destination—and our ultimate purpose is to understand the organization of cities.

While the main line of space syntax research bears little relevance to our work, there are papers about urban traffic flow that are inspired by space syntax (e.g. [1,3,4]). To our understanding, these papers follow the logic that even though the original assumptions of space syntax are violated, the derived methods still work for urban traffic. Thus, they modify or generalize space syntax methods—axial maps, integration and depth distance measures—to handle vehicular traffic, while the methods’ use for mechanistic understanding is subordinated. On the other hand, they do represent an approach to thinking about people in urban space that deserves to be mentioned in our manuscript. In principle, we can include this full discussion in the main manuscript, but in the interest of direct relevance, space, and readability, we restrict ourselves to including a reference to the most relevant paper [1] (which also contains further references to the field). This is included in the second paragraph of the manuscript.

A number of studies have conducted research on the empirical factors behind the choice of routes [23—27], yet comparatively little effort has been devoted to their geometric properties, that is their morphology.

- [1] JL Patterson, 2016, Traffic modelling in cities – Validation of space syntax at an urban scale, *Indoor and Built Environment* 25:7, 1163-1178.
- [2] B Hillier and J Hanson, 1984, *The Social Logic of Space*, Cambridge: Cambridge University Press.
- [3] X Zheng, L Zhao, M Fu and S Wang, 2008, Extension and application of space syntax: A case study of urban traffic network optimizing in Beijing. 2008 Workshop on Power Electronics and Intelligent Transportation System, pp. 291–295.
- [4] M Giannopoulou, Y Roukounis and V Stefanis, 2012, Traffic network and the urban environment: An adapted space syntax approach. *Procedia – Social and Behavioral Sciences* 48, pp. 1887–1896.

The abstract also refers to ‘diversity of road hierarchies’ but the main body of the paper does not directly refer to road hierarchy (another sub-field whose literature is by-passed), other than the acknowledgement that some routes will enable faster speeds than others. There is more direct treatment in the Supplementary Material, but this is not fully explained (whether in terms of speed, capacity, transport mode, or ‘diversity of road hierarchies’) and this would seem to deserve more discussion, to allow more confidence in the meaning and significance of the results.

Indeed, and we are puzzled by this omission, given that we provide substantial information on road hierarchies in Figs. S2-S4. We now directly quote this in the text in proper context.

In page 6 second paragraph:

The roads in our dataset are organized in a hierarchical fashion consisting of motorways, primary and trunk roads at the top of the hierarchy and residential and service roads being at the lower levels (Fig. S2). The shortest paths considered thus far are primarily composed of secondary and residential roads (Fig. S3); conversely the fastest routes tend to use a smaller subset of the overall network, primarily motorways that are in general major highways with physical divisions separating flows in opposite directions such as freeways and Autobahns (Fig S4). This heterogeneity in the road capacity is bound to introduce differences in the inness profiles of shortest and fastest routes. Reflecting this, one sees an inward bias for the fast routes emerging around 10km, but markedly less pronounced than seen for shortest routes, although the qualitative trend of increasing inward bias with r is maintained. The lower inward bias of faster routes can be explained by the fact that the motorways are typically located in the periphery of cities. Additionally, the angular range of the observed inward bias is lower than that seen for shortest paths and the fluctuations are significantly larger. This is indicative of the heterogeneous spatial distribution of velocity profiles in the primary roads across cities (due to varying levels of infrastructure), coupled with the fact that they are in general longer than secondary roads.

Page 9, first paragraph:

Having examined the properties of average directional biases across urban areas, we now turn our attention to the patterns in individual cities. Indeed, as the fluctuations in Fig. 3A show, there is variability in the inness pattern across cities, reflecting the differences in the level of road hierarchies and organization. The composition of the shortest and fastest routes in terms of different road hierarchies varies significantly from city to city. For example, in some cities (Atlanta, Houston, Madrid) the fastest routes tend to be through motorways, whereas in others (Luanda, Kolkata and Pune) these routes tend to use more primary roads (Fig S4). Such differences are likely to affect the inness profiles of those cities...

And interspersed throughout the text where relevant.

Finally, in the discussion it is claimed that 'More than the physical layout, it is the sampling of street networks that serves as a true fingerprint...' – however it is not clear what 'true fingerprint' means, in an arena when surely different metrics tell us different things – all 'true' but with different meaning or significance – about a complex system. And while the paper claims to be 'revealing the morphology of connectivity' it would seem more appropriate to say it is revealing some particular aspect(s) of the morphology of connectivity;

Agreed, the discussion has been substantially revised.

The claim that the study provides 'evidence of this evolution' (of spatial and functional expansion) seems overstated since the study does not track such evolution, and the claim seems to be no more than one can read into the map of any city some aspects of its historical development.

This is a valid point, we were indeed guilty of taking the argument too far. Based on our latest results, it seems that inness contains information on spatial and geographic characteristics as well as aspects of infrastructure and connections with composite socioeconomic indicators. Consequently, while it appears to be a novel and parsimonious indicator of various aspects of a city's development, and organization in terms of its streets, it is problematic to correlate this with evolution of cities. To say something substantial, one would need detailed longitudinal data on a variety of cities from across the world, to make a clear connection between age, development and evolution. Given that we don't have such data, we feel it best to drop this aspect of our discussion. We believe part of the problem is our liberal (and unclear) use of the word "mature" in the earlier version of the text, giving the impression that we are commenting upon the history and age of the city. We have now excised all such ambiguous words from the manuscript and have made the connection between inness and our measured metrics clear. Additionally, we have significantly toned down the discussion on Central Place and functional evolution stating instead that while it qualitatively agrees with certain elements of the theory, pending more data, making the leap between the clusters the cities lie in and their stage of evolution (beyond its street infrastructure) is not justified.

In summary, we are grateful for the referee's suggestions and valid criticism, the addressal of which we believe has significantly improved the quality and accuracy of the manuscript. We hope we have been able to address most of the concerns and look forward to a positive appraisal of the new version of the manuscript.

REVIEWERS' COMMENTS:

Reviewer #1 (Remarks to the Author):

The authors have performed a deep revision of the manuscript providing better results in the classification of cities and in the relation of the inness with socioeconomic parameters. The results are now appealing and of wide interest, the manuscript has significantly improved. I recommend thus its publication.

Reviewer #2 (Remarks to the Author):

The manuscript is significantly improved and the authors are thanked for attending to those improvements, to make what should now be a stronger paper. In particular, the additional work comparing with other indicators (e.g. section S2); and toning down claims about what the paper does.

There are yet a small number of points recommended for attention, that could help strengthen the paper or improve its reception, that are outlined below.

* * *

Page 1 (abstract) “relatively limited attention has been devoted to the interplay between street structure and its functional usage.” This statement while defensible is in danger of being misleading since there is ample work in this area though the authors do not cite it. As such this statement is unhelpful and if anything draws attention to the lack of citation of previous works that have in fact done this. Therefore finding an alternative wording is suggested.

Page 2 “comparatively little effort...” Again, this wording is not helpful and if anything draws attention to the lack of citation of literature (geography, morphology, urban and transport studies, etc) that does indeed address geometry and morphology.

Page 3 'hidden attractive forces' seems unnecessarily mysterious or presumptuous without further explanation. Geographers, historians and other urban analysts have studied many influences on urban spatial organisation for many years and the existence of 'hidden' forces unknown to geography or history seems not sufficiently justified here without further demonstration.

Page 11 'There are two possible causes' – it would perhaps be better to say 'at least two'.

Page 13 The word 'remarkably' seems rather strong without further explanation/justification.

Page 15. The concluding paragraph is unnecessarily weak. It seems to focus on the merit that a metric 'encodes information on infrastructural organization' but this of itself does not seem particularly remarkable, as that could be said of many existing network indicators. The next claim is that 'augmenting our analysis.... opens up a promising direction'; however arguably it is not the metric of itself that 'opens up' this area for research, but this is already an ongoing field area of research which may yet be enriched by the addition of an additional metric.

It could therefore be more powerful to say something more directly relating to the broader value of the paper, such as (roughly) along the lines that the paper sheds new light on a number of aspects and their relationships (e.g. inness plus road hierarchy plus fast versus direct routes plus centrality/ polycentricity plus geographical accessibility, stage of development, etc) which have been computed here for a large set of cities across the world, and which invite further more detailed and integrated analysis and interpretation with respect to existing geographical, historical, morphological and transport planning studies (etc.), for individual cities and their development trajectories.

This would reduce the dependence on one metric as an indication or explanation of anything, and draw more attention to the wider range of factors and relationships studied, that have probably not been fully explored or understood by previous researchers. Also, the implication here is that we don't yet have a full understanding of all that is implied by the data within the paper nor in previous scholarship on corresponding topics, but each needs the other to help complete our understanding.

The addition of Section S2 in particular is very welcome and indeed Figure S1 looks significant and would seem to deserve to go in the main body of the paper (although different authors and journals may have different preferences and traditions here).

Furthermore, the authors are invited to consider including part of their discussion about the value of inness over betweenness centrality (in the rebuttal [...cast doubt on the use of betweenness

centrality' and 'By contrast, inness...such a simple measure'] somewhere explicitly within the paper (whether in the main text or supplementary material)

* * *

Supplementary notes on the rebuttal document [for information]

The authors comment 'This is somewhat speculative'. Nevertheless, the referee is aware of studies and existing indicators – including works not considered in this paper – that match the point intended.

The referee is aware of different opinions on space syntax but there is no advantage in getting drawn into a discussion about its merits or relevance here. It was simply one example of a whole (sub)field of research that does in fact study the relation between street networks, urban spatial structure, urban function, movement, and urban development over time. This is to say nothing of wider literatures of urban geography, urban morphology, transport studies, road hierarchy, etc.

It was not intended here to insist on the inclusion of any particular space syntax (or any other) literature – that is left to the authors; however the paper should not leave the impression that such studies already linking network structure to function (etc) don't exist or are simply negligible. Hence the suggestions for the modifications indicated above.

Reviewer #1 (Remarks to the Author):

The authors have performed a deep revision of the manuscript providing better results in the classification of cities and in the relation of the inness with socioeconomic parameters. The results are now appealing and of wide interest, the manuscript has significantly improved. I recommend thus its publication.

Many thanks to the referee for the great reviews. The manuscript has benefited much from the suggestions.

Reviewer #2 (Remarks to the Author):

The manuscript is significantly improved and the authors are thanked for attending to those improvements, to make what should now be a stronger paper. In particular, the additional work comparing with other indicators (e.g. section S2); and toning down claims about what the paper does.

A special note of thanks to the referee. Too often reviews are rather superficial; addressing the referee's comments and constructive criticism, on the other hand, played a major role in improving the manuscript. We are in the referee's debt.

Page 1 (abstract) "relatively limited attention has been devoted to the interplay between street structure and its functional usage." This statement while defensible is in danger of being misleading since there is ample work in this area though the authors do not cite it. As such this statement is unhelpful and if anything draws attention to the lack of citation of previous works that have in fact done this. Therefore finding an alternative wording is suggested.

The abstract has been shortened and this fragment has been removed.

Page 2 "comparatively little effort..." Again, this wording is not helpful and if anything draws attention to the lack of citation of literature (geography, morphology, urban and transport studies, etc) that does indeed address geometry and morphology.

This sentence has been removed.

Page 3 'hidden attractive forces' seems unnecessarily mysterious or presumptuous without further explanation. Geographers, historians and other urban analysts have studied many influences on urban spatial organization for many years and the existence of 'hidden' forces unknown to geography or history seems not sufficiently justified here without further demonstration.

This has been changed to:

Fig.1 illustrates the forces related to a city's morphological patterns, shaping, and shaped by infrastructural and socioeconomic layouts.

Page 11 'There are two possible causes' – it would perhaps be better to say 'at least two'.

Done.

Page 13 The word 'remarkably' seems rather strong without further explanation/justification.

The word has been removed.

Page 15. The concluding paragraph is unnecessarily weak. It seems to focus on the merit that a metric 'encodes information on infrastructural organization' but this of itself does not seem particularly remarkable, as that could be said of many existing network indicators. The next claim is that 'augmenting our analysis.... opens up a promising direction'; however arguably it is not the metric of itself that 'opens up' this area for research, but this is already an ongoing field area of research which may yet be enriched by the addition of an additional metric.

It could therefore be more powerful to say something more directly relating to the broader value of the paper, such as (roughly) along the lines that the paper sheds new light on a number of aspects and their relationships (e.g. inness plus road hierarchy plus fast versus direct routes plus centrality/ polycentricity plus geographical accessibility, stage of development, etc) which have been computed here for a large set of cities across the world, and which invite further more detailed and integrated analysis and interpretation with respect to existing geographical, historical, morphological and transport planning studies (etc.), for individual cities and their development trajectories.

This is a fantastic suggestion! We have incorporated this into a modified concluding paragraph:

In summary, it appears that inness as a simple metric encodes a surprising amount of geometric, infrastructural and socio-economic information of cities. Indeed, the observed connections between inness, road hierarchies, differences in fast and direct routes, geographical accessibility, centrality in cities, socio-economic indicators and developmental stage, in combination, provide a rather comprehensive picture of urban organization. The results presented here, invite more integrated analysis and interpretation

with respect to existing geographical, historical, morphological and transport-planning studies for individual cities and their development trajectories.

The addition of Section S2 in particular is very welcome and indeed Figure S1 looks significant and would seem to deserve to go in the main body of the paper (although different authors and journals may have different preferences and traditions here). Furthermore, the authors are invited to consider including part of their discussion about the value of inness over betweenness centrality (in the rebuttal [...cast doubt on the use of betweenness centrality' and 'By contrast, inness...such a simple measure']) somewhere explicitly within the paper (whether in the main text or supplementary material)

We thank the referee for prompting us to include this analysis. In principle, it is a great suggestion to include this in the main manuscript. However, we find ourselves at the word/content limit allowed by the journal. Consequently we are compelled to keep the figure in the SI. We did however augment our discussion on the comparison between inness and networks.

On Page 4, final paragraph:

The inness of a node in the road network is a result of aggregating characteristics of all possible routes that pass through that point. Indeed, it reflects the structure of the network since it is a metric influenced by topology and network connectivity. However, in addition to this, it captures geometric aspects of the network since it is a measure based on the curvature of the roads along a route and encodes structural information at both the global and local scales. In that sense, one can consider it to contain elements of various standard structural network metrics (see~\ref{si:sec:network} for details and comparison with a series of network metrics). We note, that such metrics---particularly those that are global measures such as the betweenness centrality---have been previously used to classify cities~\cite{Kirkley_2017}. On the other hand, as we will demonstrate, the inness, in addition to being a relatively simple measure, encodes geometric, infrastructural, geographical and socioeconomic aspects of urban systems.